# Current and Future Treatments in the Fight against Non-Alcoholic Fatty Liver Disease

**DOI:** 10.3390/cancers12071714

**Published:** 2020-06-28

**Authors:** Benoit Smeuninx, Ebru Boslem, Mark A. Febbraio

**Affiliations:** Cellular & Molecular Metabolism Laboratory, Monash Institute of Pharmacological Sciences, Monash University, Parkville, VIC 3052, Australia; benoit.smeuninx@monash.edu (B.S.); ebru.boslem@monash.edu (E.B.)

**Keywords:** NAFLD, NASH, hepatocellular carcinoma, treatments

## Abstract

Obesity is recognised as a risk factor for many types of cancers, in particular hepatocellular carcinoma (HCC). A critical factor in the development of HCC from non-alcoholic fatty liver disease (NAFLD) is the presence of non-alcoholic steatohepatitis (NASH). Therapies aimed at NASH to reduce the risk of HCC are sparse and largely unsuccessful. Lifestyle modifications such as diet and regular exercise have poor adherence. Moreover, current pharmacological treatments such as pioglitazone and vitamin E have limited effects on fibrosis, a key risk factor in HCC progression. As NAFLD is becoming more prevalent in developed countries due to rising rates of obesity, a need for directed treatment is imperative. Numerous novel therapies including PPAR agonists, anti-fibrotic therapies and agents targeting inflammation, oxidative stress and the gut-liver axis are currently in development, with the aim of targeting key processes in the progression of NASH and HCC. Here, we critically evaluate literature on the aetiology of NAFLD-related HCC, and explore the potential treatment options for NASH and HCC.

## 1. Introduction

Non-alcoholic fatty liver disease (NAFLD) encompasses a spectrum of histological liver abnormalities ranging from bland steatosis to liver cirrhosis, which could lead to the development of hepatocellular carcinoma (HCC) [1]. Bland steatosis, defined by the excessive accumulation of hepatic triglycerides, does not lead to liver damage itself, but lays the foundation for the development of the more aggressive non-alcoholic steatohepatitis (NASH). This disease spectrum has led to the formulation of the double- or multiple-hit hypothesis; where steatosis must be accompanied by at least one other factor such as inflammation, oxidative stress, or endoplasmic reticulum (ER) stress to trigger a necro-inflammatory response catalysing the progression from NAFLD towards NASH [2]. How NASH then evolves to HCC is completely unclear, and mechanistic data obtained from animal models has not translated to human disease [3]. Elucidating these factors catalysing NASH to HCC progression is crucial as HCC is one of the most lethal and fastest rising cancers worldwide [4].

An estimated 25% of the global population suffers from NAFLD, with around 15% of those exhibiting signs of NASH. The prevalence of NAFLD is closely linked to obesity and metabolic disease, and is hence widespread in Western countries [4,5]. Nevertheless, evidence is emerging for the presence of NAFLD in lean individuals. The mechanisms underlying NALFD development in the absence of obesity are not well-understood but may include metabolic, genetic and environmental factors [6]. By definition the prevalence of NAFLD excludes individuals consuming alcohol in excess of 20–30 g·day^−1^. This does not, however, equate to a total abstention from alcohol [7]. On the contrary, a significant fraction of NAFLD patients are moderate to heavy alcohol consumers. Furthermore, research has suggested binge drinking to serve as a second-hit, besides steatosis, in the development of NAFLD [8]. Therefore, it is challenging to tease out the intricate differences between NAFLD and AFLD. The direct clinical and added indirect annual social costs associated with NAFLD are enormous, and estimated, in the U.S. alone, at USD 103 and USD 292 billion, respectively [9]. This highlights the urgent need to develop effective NAFLD screening tools and therapies.

Early NAFLD diagnosis is crucial to manage disease progression and assure patients can benefit from the latest most effective therapies. To date, the gold standard to confirm NAFLD presence is with a liver biopsy. Based on the biopsy, a NAFLD activity score (NAS) is calculated and considers four important histological features, i.e., steatosis (0–3), hepatocellular ballooning (0–2), lobular inflammation (0–3) and fibrosis stage (0–4). Each feature is given a score with the total sum of individual scores indicating the likeliness of NASH (NAS ≥ 5) [10]. However, the invasive and costly nature of the liver biopsy procedure has prompted its selected use for high-risk patients only [11]. Furthermore, concerns have been raised regarding the obtainment of non-representative biopsy samples as histological lesions of NASH are unevenly distributed throughout the liver parenchyma [12]. This has prompted the development of non-invasive imaging techniques, or discovery of circulatory biomarkers of liver damage to determine the degree of liver fibrosis [11].

The widespread prevalence of NAFLD with its comorbidities emphasises the urgent need for effective treatments. A better understanding of the mechanisms underpinning NAFLD is essential for the discovery of novel treatments. Furthermore, pinpointing cost effective, easy-to-assess and reliable biomarkers of NAFLD development would allow us to treat patients before irreversible liver damage has occurred. To date, several promising therapeutic targets have been unveiled using animal models of NAFLD and NASH [3]. However, when pharmacological agents are put to the test in human clinical NAFLD trials, results mostly do not live up to the expectation. Furthermore, the heterogeneous nature of NAFLD makes it challenging to find the “holy grail” of treatments. Here, we provide a broad overview of present and future therapies aimed at attenuating, or even reversing, NAFLD progression.

## 2. Treatments

The rapidly growing number of individuals being diagnosed with NAFLD and the associated burden on national health care costs has led to an increased focus on developing novel therapies to prevent, treat, or cure the disease [9]. Whilst NAFLD is a liver-specific condition, both intra- and extra-hepatic factors play a role in its development and can, hence, be targeted for treatment purposes [13]. To date, most NAFLD therapies are aimed at ameliorating one pathophysiological risk factor. However, the extremely complex nature of NAFLD, where no one clearly identifiable agent underlies the disease, makes it near impossible to resolve the condition by targeting one risk factor. Consequently, most treatments will indirectly influence a multitude of underlying mechanisms. Here, we review past, current and pipeline treatments according to their mode of action. First, treatments targeted at the main characteristics driving NAFLD progression are discussed (steatosis, inflammation and fibrosis), followed by treatments aimed at the gut microbiome in order to ameliorate the gut-liver interaction. It is important to note that the NAFLD treatment landscape is far from static with novel pharmacological agents being developed continuously to meet global demand. The below summary is, therefore, only a snapshot in time. Figure 1 provides a graphical overview of treatments impacting NAFLD.

### 2.1. Hallmark Characteristics Driving NAFLD

#### 2.1.1. Steatosis

Steatosis, the first hallmark characteristic of NAFLD, is ultimately the consequence of an imbalance in hepatic lipid turnover where synthesis rates chronically exceed breakdown rates [14]. Two major mechanisms contribute to elevated synthesis rates: specifically, (i) increased hepatic fatty acid delivery due to an increase in adipose triglyceride lipolysis—a process strongly dictated by insulin signalling [15]; and (ii) de novo *lipogenesis* (DNL)—a process where glucose and fructose are enzymatically converted by acetyl-CoA carboxylase (ACC) into lipids [16]. Lipid breakdown rates are predominantly dictated by mitochondrial lipid oxidation and VLDL secretion and export [15,17,18]. Treatments tackling steatosis are, hence, aimed at either reducing lipid accumulation or increasing fatty acid removal.

Inactivity and excessive food and alcohol intake are the main factors contributing to hepatic steatosis [19,20]. It is, therefore, unsurprising that calorie restriction and/or exercise are the most effective preventative treatments to date. Indeed, restricting calorie intake decreases lipid substrate delivery, whilst increasing physical activity mobilizes fatty acids out of the liver. These two mechanisms combined will, ultimately, reduce steatosis [21,22]. In this regard, an overall weight loss of 5% has repeatedly been shown to reduce steatosis and improve NAFLD [23], whilst a weight loss of 10% will likely lead to NASH resolution and an improvement of at least 1 stage in fibrosis score [24]. Interestingly, substituting a high-fructose Western diet with a typical Mediterranean diet can improve NAFLD in the absence of weight reduction [25]. Long-term implementation of, and adherence to, these life-style changes is challenging and not all patients are up to the commitment [26]. The observation of patients regaining weight after initial weight loss is, therefore, not uncommon, and is biologically hardwired via the hyperstimulation of the dopaminergic reward centres in the hypothalamus that drive increased hunger over satiety signalling, and the associated increase in feeding behaviour seen in obesity [27]. However, it is important to emphasise the lasting benefits of weight loss on liver steatosis and insulin sensitivity even when weight has been regained [28]. While weight loss is important as a lifestyle treatment intervention, regular aerobic exercise alone can be effective in treating not only NASH [29], but also ASH, at least in studies of preclinical models [30].

The difficulty in maintaining adequate weight loss and physical activity has spurred the development of pharmacological treatments inhibiting fatty acid accumulation in the liver. The majority of these pharmacological agents do so by improving insulin sensitivity and glycaemic control or by decreasing DNL. Peroxisome proliferator-activator receptors (PPARs) are nuclear receptors ubiquitously expressed in liver, heart, adipose tissue, and skeletal muscle and consist of three distinct receptors: α, δ and γ. All three family members play an important role in lipid metabolism and energy homeostasis. In this regard, PPARα agonists, such as fibrates, induce the expression of genes involved in fatty acid β-oxidation and insulin sensitization whilst downregulating nuclear factor-κB (NF-κB) [31]. PPARδ agonists reduce fatty acid uptake and alter glucose homeostasis [32]. Finally, PPARγ agonists improve insulin sensitivity and lower blood glucose [33]. These distinct characteristics render PPAR agonists prime candidates to reduce steatosis in order to ameliorate NAFLD progression.

##### PPARα Agonists

Two decades ago, gemfibrozil was one of the first PPARα agonists used to treat NASH patients. Gemfibrozil lowered aspartate (AST) and alanine aminotransferases (ALT), reduced peripheral fatty acid mobilization, and induced hepatic lipid clearance [34]. Since then, several fibrates have been developed primarily to treat dyslipidaemia, but have been adopted to ameliorate steatosis. In this regard, fenofibrate was evaluated in a 48-week trial in patients with biopsy proven NAFLD [35]. Whilst improving metabolic syndrome-related parameters, fenofibrate did not affect liver histology. Clofibrate, another PPARα agonist, was evaluated as a potential treatment for NASH patients, but was unable to improve ALT, AST, steatosis, inflammation, and fibrosis [36]. More recently, the effects of omega-3 poly-unsaturated fatty acids (n-3 PUFA) in NASH patients have been investigated. Omega-3 PUFA induce PPARα activation, leading to increased mitochondrial fatty acid oxidation and inhibition of sterol regulatory element-binding protein 1 (SREB-1C); a key transcription factor upregulating the expression of enzymes catalysing DNL [37]. The efficacy of n-3 PUFA to treat NAFLD was evaluated in the Welcome study [38]. Here, patients with NAFLD were treated for 15-18 months with either a placebo or Omacor, a purified n-3 PUFA supplement containing docosahexaenoic acid (DHA) and eicosapentaenoic acid (EPA). Results did not reveal any improvements in fibrosis score, but did show a linear decrease in liver fat percentage. Further evaluation of Omacor’s effectiveness to treat NASH has been undertaken in a phase 3 clinical trial. However, results are yet to be disclosed (NCT01277237).

##### PPARδ Agonists

The second isoform of the PPAR family, PPARδ, is highly expressed in hepatocytes, Kupffer cells and hepatic stellate cells, suggesting it has an additive function in fibrosis and inflammation besides its role in steatosis [39]. Activation through its agonist GW501516 reduced serum triglycerides and prevented a decrease in HDL-c and apoA-1 levels in sedentary individuals [40]. However, safety concerns associated with GW501516 led to the search for novel PPARδ agonists. As a result, Seladelpar (MBX-8025) was developed and shown to be capable of improving serum lipid profiles and reducing liver enzyme levels in dyslipidemic patients [41]. Furthermore, in diabetic obese mice, seladelpar improved insulin sensitivity and decreased hepatic lipotoxicity [42]. Nevertheless, a phase 2 clinical trial in NASH patients evaluating seladelpar had to be suspended due to unexpected histological findings (NCT03551522). So far, no PPARδ agonists are in clinical use. However, the search for highly specific agonists is ongoing and could yield promising results [43].

##### PPARγ Agonists

The PPARγ ligands, typically referred to as thiazolidinediones (TZDs), are designed to reduce hepatic steatosis by sensitizing the liver to insulin. Mainly used to treat diabetes, recent interest has surged in their ability to reduce steatosis and attenuate NAFLD [44]. However, caution is warranted as increases in body mass due to water retention and congestive heart failure are reported side-effects of TZDs use [45].

The ability of pioglitazone, a commonly used PPARγ agonist, to resolve NASH has been evaluated alongside vitamin E in the PIVENS phase 3 trial [46]. A total of 247 patients were included and randomly assigned to either receive 30 mg Pioglitazone daily, 800 IU Vit E daily, or a matching placebo for 96 weeks. Results revealed a reduction in steatosis and lobular inflammation with pioglitazone and vitamin E use, but no change in fibrosis [46]. When long-term pioglitazone treatment was combined with a daily 500 kcal dietary deficit, NASH resolution was achieved in 51% of prediabetic and diabetic patients [47]. Nevertheless, the negative side-effects associated with TZD use have spurred the development of novel PPARγ agonists such as lobeglitazone. A 24-week lobeglitazone treatment intervention improved glycaemic, liver and lipid profiles in type 2 diabetes (T2D) patients with NAFLD, and was accompanied by a more modest weight gain as opposed to pioglitazone [48]. Further clinical trials are warranted to assess lobeglitazone’s effectiveness in treating NASH.

##### Dual PPAR Agonists

Considering the potential ability of individual PPAR ligands to slow NAFLD progression, combining multiple receptor agonists may provide better disease management. Ligands activating both PPARα and δ have shown promising results so far. Elafibranor (GFT505), a dual PPARα/δ agonist, has shown efficacy in rodent models of NAFLD and fibrosis [49]. When evaluated in a phase 2a clinical trial, elafibranor improved plasma lipids, hepatic insulin resistance, glucose homeostasis and reduced liver inflammation in pre- and diabetic patients [50,51]. These favourable results instigated the evaluation of elafibranor’s safety and efficacy in patients with NASH [52]. Accordingly, daily administration of 120 mg for one year was well tolerated and did not induce weight gain or cardiac events in NASH patients without cirrhosis. Although, results failed to show a significant difference between the 120 mg daily elafibranor and placebo group in the intention-to-treat analysis, NASH was resolved to a higher extent in the treatment compared with placebo group (19% vs. 12%, respectively) [52]. Recently, elafibranor failed to improve NASH resolution (primary outcome) and fibrosis (secondary outcome) in NASH patients over a placebo treatment in the RESOLVE-IT phase 3 clinical trial (NCT02704403).

In addition to α/δ receptor ligands, the efficacy of α/γ ligands has been evaluated to treat NAFLD. Saroglitazar, a dual PPARα/γ agonist, was successfully tested in a NASH mouse model [53]. When administered in a dose-dependent fashion, analyses revealed that full PPARα activation coincided with only partial PPARγ activation. Due to this unique property, saroglitazar was able to negate the PPARγ-associated side effects whilst still showing lipid-lowering and insulin sensitizing effects [31]. Currently, a phase 2 clinical trial is evaluating the efficacy of saroglitazar in NASH patients (NCT03863574), but the results of this clinical trial are not yet available.

##### PPAR-pan Agonists

The ubiquitous activation of all three PPARs is achieved by PPAR-pan agonists and aims to target a larger array of mechanisms underpinning NAFLD as opposed to single or dual PPAR agonists. This led to the discovery of the PPARpan activator IVA337 (lanifibranor), a moderately active and well-balanced PPARpan agonist with great potential for NAFLD treatment [54]. When tested in several rodent models of NASH, the new-generation PPARpan agonist IVA337 had preventative effects on fibrosis and decreased inflammation, steatosis and ballooning [55]. Lanifibranor as a treatment for NASH is under investigation in the NATIVE phase 2b clinical trial and has recently received FDA fast-track approval (NCT03459079). Nevertheless, several safety concerns, such as carcinogenesis, have been raised when previous PPARpan agonists have been evaluated. This has led to the discontinuation of GW677964, DRL-11605, GW25019 and netoglitazone [56,57,58]. Despite the attractive features that PPARpan agonists exhibit, more clinical trials are needed to establish their safety and effectiveness in patient populations.

##### Non-PPAR Treatments

Apart from PPAR agonists, other pharmacological agents are capable of altering substrate delivery and utilisation in order to prevent NAFLD progression. A potent regulator of substrate utilization is insulin, and as NAFLD is closely linked to insulin resistance, the ability of direct oral insulin administration to attenuate hepatic steatosis is currently under investigation in a phase 2 pilot study (NCT02653300).

Incretin hormones, such as glucagon-like peptide 1 (GLP1), improve hyperglycaemia via insulin secretion when glucose is orally consumed [59]. As such, the pharmacological GLP1 agonist, liraglutide, was produced using recombinant DNA technology and has a 97% common amino acid sequence homology to endogenously produced human GLP1 [60]. Daily subcutaneous injection of liraglutide for 24 weeks in glucose intolerant NASH patients significantly improved liver function and histological features [61]. Similar results were found for the GLP1 agonist semaglitude, which reduced circulatory levels of ALT and high-sensitivity C-reactive protein in subjects at risk for NAFLD [62]. The effectiveness of liraglutide and semaglutide are currently being examined in phase 2 clinical trials (NCT01237119 and NCT03987451).

A different, but promising, strategy to counteract NAFLD targets stearoyl coenzyme A desaturase 1 (SCD1) activity. SCD1 inhibition decreases fatty acid synthesis and increases β-oxidation, leading to decreased levels of obesity, fatty liver and insulin resistance [63]. In this regard, aramchol, a synthetic lipid obtained by conjugating cholic acid and arachidic acid through a stable amide bond, has been shown to inhibit SCD1 activity in vitro [64]. Aramchol administration exerted antifibrotic effects in mice fed a methione choline deficient diet [65], and decreased liver fat content by 12.5% over a 3-month treatment period in patients with NAFLD (contrary to a 6.4% increase in the placebo control treatment) [66]. Aramchol is currently being investigated in a phase 3/4 clinical trial (NCT04104321).

An additional pathway through which NAFLD can be attenuated is by inhibiting the sodium-glucose co-transporter 2 (SGLT2). Originally designed to treat diabetes, SGLT2 inhibitors increase urinary glucose excretion in order to reduce hyperglycaemia in an insulin-independent fashion [67]. Several SGLT2 inhibitors, including empagliflozin [68], ipragliflozin [69], canagliflozin [70], and luseogliflozin [71], improved steatosis and NASH development in rodent models of NAFLD. Translation of these findings in human clinical trials has yielded positive results. As such, dapagliflozin improved liver steatosis in NAFLD patients with T2D and improved fibrosis in those with significant liver fibrosis. However, dapagliflozin treatment resulted in weight loss and reductions in visceral adipose tissue, potentially obscuring its effects on steatosis and fibrosis [72]. Dapagliflozin has since been evaluated in the EFFECT-II study, a phase 2 clinical trial, where it reduced all measured biomarkers of hepatocyte injury [73]. A phase 3 clinical trial is currently being conducted to look at dapagliflozin’s efficacy and action in patients with NASH (NCT03723252).

Recently, work from our own group evaluated the efficacy of activation of a chimeric protein IC7Fc for the treatment of NAFLD, insulin resistance and T2D in preclinical models and in non-human primates. IC7Fc activates the IL-6 signalling cascade, but in a completely unique manner and leads to weight loss, decreased food intake, and an increased incretin response [74]. Moreover, the peptide was shown to have efficacy and safety in non-human primates. Human clinical trials with ICFc or next generation molecules are planned.

#### 2.1.2. Inflammation

Inflammation is the second hallmark characteristic fuelling the progression from bland steatosis to NASH [3]. Reducing inflammation is pivotal as sustained inflammation drives fibrosis, ultimately leading to liver cirrhosis [1]. A myriad of intra- and extra-hepatic factors can trigger inflammation including hepatocellular stress, oxidative stress, and dietary and lifestyle behaviours [17,75]. Therefore, therapeutics that lower the inflammasome will attenuate NAFLD progression.

Most treatments that improve NAFLD directly target one of several underlying mechanisms. Nevertheless, concomitant clinical improvements of other mechanisms are often observed as a result. As such, PPAR agonists and Farnesoid X receptor (FXR) ligands directly reduce steatosis and fibrosis, respectively, but will also affect inflammation indirectly [32,76].

Pentoxifylline, a methylxanthine drug, has been shown to inhibit TNF, a key player in the inflammatory response [77]. This specific trait renders pentoxifylline an appealing anti-NASH drug candidate. When put to the test in a phase 2 clinical trial, pentoxifylline improved steatosis, lobular inflammation and, although not significant, fibrosis in patients with NASH. No effect on hepatocellular ballooning was observed [78]. Nevertheless, pentoxifylline failed to meet the hypothesised proof of concept as circulatory levels of TNF remained unchanged. In a separate phase 2 trial, a 12-month pentoxifylline intervention did not improve ALT levels (NCT00267670). However, improvements in hepatic histological activity were observed when an identical dose of pentoxifylline was accompanied by lifestyle changes [79]. Results concerning pentoxifylline seem promising, but cannot be overinterpreted, as blinded randomized controlled phase 3 trials are yet to be conducted [80].

The role of vitamin D, and especially the biologically active 1,25-dihydroxyvitamin D, is key to several aspects of human metabolism [81]. Deficiencies are associated with the development of T2D [82], obesity [83] and the metabolic syndrome [84]. More recently, vitamin D deficiency has been shown to play a role in inflammation, oxidative stress, toll-like receptor activation and NAFLD [85]. Indeed, vitamin D deficiency is associated with an increased risk for NASH development in NAFLD individuals through activation of the mitogen-activated protein kinase (MAPK) and NF-κB pathways [86]. Vitamin D supplementation is, hence, an attractive treatment for NAFLD and its progression towards NASH. In a phase 2 clinical trial, 2100 IU of vitamin D for 48 weeks improved serum ALT levels and histology-proven NASH in hypovitaminosis-D patients with NASH [87]. The frequent manifestation of vitamin D deficiency in NAFLD patient groups necessitates prospective longitudinal studies to determine optimal dosing strategies [88].

More recently, a role for the medium chain free fatty acid receptor G protein-coupled receptor 84 (GPR84) has been proposed in the development of NAFLD. GPR84 is induced in immune cells under inflammatory conditions promoting phagocytosis and activation of murine and human macrophages [89]. In livers of NAFLD patients GPR84 expression was found to be increased, whilst pharmacological inhibition of GPR84 reduced inflammation and fibrosis in three different NASH mouse models [90]. GPR84 could, hence, be a promising novel therapeutic target to ameliorate NAFLD.

Finally, the gut microbiome is pivotal to overall health and, in particular, to metabolism and immunity [91]. Antibiotics that target the gut-liver axis might, hence, decrease hepatocellular inflammation by inducing favourable effects on the gut microbiome and are discussed below [92].

Attenuating inflammation is key for successful NAFLD treatment. Whilst significant strides towards successful anti-inflammatory therapies have been made, confirmation in randomized controlled clinical trials is needed.

#### 2.1.3. Fibrosis

Patients diagnosed with NAFLD can develop NASH with or without fibrosis. Previously, fibrosis stage was included in the definition of NASH but has more recently been omitted. As fibrosis severity (or stage), rather than NASH, is predictive of mortality, it is uncertain whether NASH itself is associated with adverse outcomes [93,94]. Reducing fibrosis should, therefore, be a main objective of NASH-related therapeutics. Fibrogenesis is induced by the signalling of activated macrophages and injured hepatocytes to hepatic stellate cells, which, in turn, will develop into myofibroblasts and initiate the production of matrix proteins [95]. It is this excessive extracellular matrix production, which is not adequately broken down, that leads to the accumulation of hepatic fibrosis. The pathways promoting fibrogenesis are increasingly being unravelled, paving the way for the discovery of novel pharmacological agents.

One mechanism by which fibrosis can be attenuated, is through activation of the nuclear Farnesoid X receptor (FXR). Natural FXR activation occurs via bile acids. Once synthesised in the liver, bile acids are metabolised in the gut by bacteria and sensed by FXR residing in epithelial cells. FXR will then signal back to the liver via fibroblast growth factor 19 (FGF19) or its rodent homolog FGF15 [96]. The potent FXR activator, obeticholic acid (OCA), reduced fibrosis in mouse models of NASH [97,98]. In a human phase 2B trial, daily consumption of 25 mg OCA improved histological features of NASH. However, 23% of patients developed pruritus and increased VLDL due to OCA consumption [99]. To establish whether a lower OCA dose would reduce pruritus occurrence whilst still improving histological features of NASH, the REGENERATE trial investigated the effectiveness of a 10 and 25 mg daily OCA dose. The fibrosis improvement endpoint was achieved by 18% (*p* = 0.045) in the OCA 10 mg group and by 23 % in the OCA 25 mg group (*p* = 0.0002). Pruritus occurrence was 28 and 55% in the 10 and 25 mg OCA group, respectively [100]. Whilst the long-term effects of OCA consumption on health are still up for debate, the drug has the ability to ameliorate NASH. Accordingly, on March 6th 2020 the FDA granted breakthrough therapy designation to Intercept Pharmaceutical’s for OCA as a therapy to treat NASH with liver fibrosis. Recently, Intercept Pharmaceutical has developed INT-787, which appears to be a more specific FXR agonist than OCA, and is being evaluated in preclinical studies.

Lower circulatory FGF19 concentrations, potentially linked to a dysregulated FXR signalling, are often observed in patients with NASH and contribute towards disease progression [101]. Rodent studies have explored the therapeutic potential of FGF19, but these have been hindered by its hepatocarcinogenic properties [102]. The creation of the FGF19 variant, NGM282, prevented this problem, as it does not activate signal transducer and activator of transcription 3 (STAT3) [103]. A phase 2 clinical trial investigated the safety and effectiveness of a 3 or 6 mg daily injection of NGM282 for 12 weeks in patients with NASH. NGM282 at both doses led to a rapid and significant reduction in liver fat content, warranting further exploration of the drug’s therapeutic properties in NASH patients [104].

Similarly, activation of the bile acid receptor Takeda G protein-coupled receptor 5 (TGR5) has been shown to reduce inflammation and increase energy expenditure and oxygen consumption. TGR5 is highly expressed in cholangiocytes, Kupffer cells, CD14+ cells and the intestine. Activation of TGR5 through its agonist S-EMCA, INT-777 was found to improve liver function by modulating GLP-1 expression in obese mice [105], whilst RDX8940-induced TGR5 activation improved hepatic steatosis and insulin sensitivity in mice fed a Western diet [106]. Dual FXR and TGR5 activation through semisynthetic bile acid derivatives has shown promising results in rodent NAFLD models. In this regard, INT-767 reduced inflammation and improved liver histopathology in db/db mice with NAFLD [107] and ob/ob mice with NASH [108]. Whilst INT-767’s safety has been evaluated by Intercept Pharmaceutical in healthy volunteers, clinical trials are needed to evaluate its efficacy in patients with NAFLD.

Hepatic stellate cell activation plays a central role in fibrogenesis and, hence, direct inhibition of hepatic stellate cells could attenuate fibrogenesis [95,109]. This approach is currently under investigation in a phase 1b/2 trial using a vitamin-A coupled lipid nanoparticle containing a siRNA against heath shock protein 47 (HSP47) (NCT02227459). Folding of fibrillary collagen is chaperoned by HSP47 and silencing its expression should result in collagen misfolding and hepatic stellate cell apoptosis [110].

Finally, selonsertib, a potent and selective small molecule inhibitor of apoptosis signal-regulating kinase 1 (ASK1), successfully improved liver fibrosis score by ≥1 stage in NASH patients with stage 2 or 3 liver fibrosis [111]. Nevertheless, in two phase 3 clinical trials oral consumption of either 6 or 8 mg of selonsertib was unable to achieve the primary end goal of improving fibrosis ≥1 stage, and were hence terminated early [112].

The inverse correlation between survival and fibrosis severity necessitates the development of successful anti-fibrotic treatments. Thus far, promising preliminary results have been obtained, but long-term clinical trials are warranted in clinical NAFLD populations.

### 2.2. Gut Microbiome

The gut microbiome plays a key role in maintaining overall health. As such, recognition of its ability to affect the host’s immune system and, consequently, disease progression, has risen over the past few years. Closer examination of the gut microbiome reveals a plethora of residing organisms including, but not limited to, eukaryotic microbes, archaea, viruses and fungi [91,113]. The direct and/or indirect contribution of these microorganisms to NAFLD is steadily emerging and opens the door to several new anti-NAFLD therapies [92]. Chronic liver disease induces alterations to the gut microbiome and intestinal barrier, which results in dysbiosis and leaky gut. Disruptions in the intestinal barrier allow pro-carcinogenic agents to diffuse into the circulation and affect the liver [114]. Interestingly, high fructose consumption has been associated with liver disease and two recent studies have demonstrated a link between high fructose consumption, gut dysbiosis and liver steatosis and inflammation [115,116]. Here, we briefly discuss treatments aimed altering the gut microbiome and gut-liver axis in order to alleviate NAFLD disease progression.

#### 2.2.1. Antibiotics

The strong association between gut dysbiosis and NAFLD development poses the question whether antibiotics (ABX) can attenuate NAFLD through inducing alterations in the gut microbiome. The central role of the gut microbiome in disease has been elegantly shown in a study by Henao-Mejia et al., where the cohousing of wild-type with inflammasome-deficient mice led to hepatic steatosis and obesity in the wild type mice. However, administration of ABX abrogated the observed abnormalities [117]. The ability of ABX to alter the gut microbiome composition and decrease leaky gut in order to ameliorate NAFLD progression has since been shown in several rodent studies [118,119]. In a recent study, high fructose feeding induced NASH and HCC in mice, and was ameliorated by ABX treatment [116]. Nevertheless, caution is warranted when extrapolating these results into a human population as the total ablation of the gut microbiome would not be feasible nor would it be desirable. The development of ABX targeted at specific microbiome subpopulations that play a critical role in HCC development could potentially change the HCC research field. In this regard, norfloxacin specifically reduced gram-negative bacteria attenuating bacterial infections in cirrhotic patients [120]. Rifaximin, a minimally absorbable antibiotic acting on gram-negative microbes in the gut, ameliorated alcoholic liver disease in rodents [121] and prevented hepatic encephalopathy in patients [122]. In patients with biopsy proven NASH, twice daily oral administration of 550 mg rifaximin for 6 months significantly decreased serum ALT and AST levels. However, no effects on serum cholesterol and triglycerides were found [123]. A clinical study conducted by Cobbold and colleagues failed to observe any beneficial effects of rifaximin administration (400 mg twice daily for 6 weeks) in patients with NASH [124]. Larger clinical trials are needed to elucidate whether rifaximin alone, or in combination with other treatments, could be a viable strategy to combat NASH.

Besides reducing microbial subpopulations linked to HCC development, the microbial milieu can be modulated in order to increase antibiotic and immunotherapy susceptibility [125].

#### 2.2.2. Probiotics

Probiotics are live microbial food ingredients that, when administered in adequate amounts, benefit the host [126]. The mechanisms through which probiotics exert their positive effects are still to be fully elucidated but include an improvement in the gut microbiome, gut epithelial barrier function, and modulation of local and systemic immunity [127]. Studies in rodents have demonstrated the benefits of probiotics on hepatocarcinogenesis [128]. Specifically, the probiotic VSL#3 protected against NASH development in dextran sulphate sodium-treated ApoE^-/-^ mice [129], ob/ob mice [130], FXR knock-out mice [131], and HFD fed rats [132]. Furthermore, VSL#3 showed promising results in obese children with NASH [133]. Together, these data point towards a potential protective role of probiotic consumption, which needs to be confirmed in large-scale clinical trials. The low-cost and safe-to-consume profile of probiotics categorises them as a desirable intervention to attenuate NAFLD progression.

#### 2.2.3. Faecal Microbiota Transplantation (FMT)

Faecal microbiota transplantation (FMT) refers to the introduction of the faecal microbiome from a healthy donor into the gastrointestinal tract of a patient recipient, this can be done within or across species [134]. The technique was first implemented to combat *Clostridium Difficile* and has since been successful in alleviating obesity and T2D [135,136]. Faecal microbiome transplantation from patients with NASH into mice caused the development of NASH in the latter, showing a clear link between the microbiome and disease development [137]. However, the use of FMT is not without risks as the transfer of harmful drug-resistant organisms from the donor into the host can easily occur. This is especially of concern if the host recipient suffers from an immuno-suppressive illness [138]. Whilst FMT has many benefits, one cannot ignore its disadvantages. Moving away from FMT towards bacterial “cocktails” aimed at the disease’s specific gut microbial fingerprint would prove safer and pave the way towards individualized treatments.

#### 2.2.4. Leaky Gut

Leaky gut caused by disruptions in the tight junctions of the gut endothelial cells can lead to an outwards flux of microbe-associated molecular patterns (MAMPs) from the gut into the portal bed affecting the liver [92]. Factors contributing to leaky gut are not fully understood but include intestinal inflammation as a consequence of dysbiosis, decreased bile acid secretion, a compromised immune system, and an increased permeability of the gut-vascular barrier [92,139]. Leaky gut has been shown to contribute to HCC development via lipopolysaccharides (LPS) and its receptor TLR4. In this regard, a low-dose infusion of LPS in mice promoted HCC development [116,121]. Pharmacological agents targeting the gut permeability hold, therefore, promise in treating chronic liver disease such as NAFLD. Bile acids are potent regulators of gut permeability and bind endothelial FXR receptors, which in turn release FGF19 to act on targets in the liver. FXR agonists, such as OCA, can attenuate mucosal injury and ileal barrier permeability, leading to decreased bacterial translocation [140,141]. As discussed above OCA was recently FDA approved to treat NASH and liver fibrosis. A central role has been attributed to the G protein-coupled chemokine receptor CX3CR1 (GPR-13) in maintaining intestinal homeostasis. In this regard, CX3CR1 reduced hepatic steatosis and inflammation in both a high-fat and methionine/choline deficient diet-induced mouse model of steatohepatitis [142].

An overview of the discussed treatments and their main outcomes is presented in Table 1.

## 3. Lean NAFLD

The occurrence of NAFLD is strongly associated with obesity and metabolic disease. Consequently, the condition it is often overlooked in non-obese individuals, which could be catastrophic, as 10–20% of non-obese Americans have been diagnosed with NAFLD [143,144]. This so-called “lean NAFLD” is defined as the occurrence of NAFLD in individuals with a BMI below the ethnic-specific cut-off for being overweight (25 kg·m^2^ in Caucasian and 23 kg·m^2^ in Asian populations). Although lean individuals suffering from NAFLD appear healthier, they can display NASH-like symptoms to the same extent as their obese counterparts and are, therefore, often referred to as metabolically obese normal weight. Increasing awareness for the occurrence of NAFLD in the absence of classic metabolic risk factors is needed [145].

Several factors contribute to the pathogenesis of NAFLD in lean individuals, and range from lifestyle behaviours to genetic factors [146]. As such, the consumption foods high in either fructose [147] and/or cholesterol (Western diet) [148], is positively associated with the development of NAFLD and is independent of metabolic disease. Similarly, increased sitting time was positively correlated with the prevalence of NAFLD in individuals with a BMI < 23 kg·m^−2^ [149]. Further adding to the distinct pathophysiology of lean NAFLD are alterations in the gut microbiome with, at the genus level, elevated *Erysipelotrichaceae* UCG-003 as well as multiple bacterial genera that are part of the *Clostridiales* order [150]. Elevated total primary and secondary serum bile acid levels were observed in lean NAFLD, potentially leading to diet-induced obesity resistance through an FGF19-related increase in energy expenditure [150].

The importance of genetic factors predisposing individuals to lean NAFLD is gaining momentum. One of the first genes associated with lean NASH was the patatin-like phospholipase domain-containing protein 3 (PNPLA3) and its nonsynonymous variant rs738409 C > G (I148M), where a methionine is substituted for an isoleucine [151,152]. The I148M variant perturbs hepatic lipid homeostasis via the inhibition of glycerolipid hydrolysis, resulting in a decreased hepatic lipid outflow [153]. Other genetic variants associated with lean NAFLD include an increase in the single nucleotide polymorphisms rs12447924 and rs12597002 on the cholesterol ester transfer protein (CETP) [154], and a decrease in phosphatidylethanolamine N-methyltransferase (PEMT) [155]. In contrast, a variant of the 17β hydroxy steroid dehydrogenase 13 gene (rs72613567) was found to protect against NAFLD development [156].

To date, no specific treatments for lean-NAFLD exist. Besides lifestyle interventions targeting a healthier diet and increased physical activity, pharmacological treatments for high blood pressure, dyslipidaemia and hyperglycaemia are often prescribed. As such, the TZD pioglitazone was successful in lowering blood glucose in non-obese diabetic individuals. However, analysis was not performed in lean individuals [157]. The steady discovery of specific mechanisms and genetic fingerprints underpinning lean, as opposed to non-lean, NAFLD will allow for the development of more targeted interventions.

## 4. Conclusions

One in five individuals globally are estimated to have NAFLD. Its prevalence is strongly associated with obesity and metabolic disorders; however, evidence is mounting for its occurrence in non-obese individuals. Whilst bland steatosis itself is not harmful, it lays the foundation for the development of NASH and HCC. The latter has been touted as being among the most lethal and fastest growing cancers worldwide, emphasizing the need for effective treatments.

The complex and heterogenous nature of NAFLD challenges the quest to find the holy grail of treatments. So far treatments are generally aimed at directly ameliorating either one of the hallmark characteristics driving NAFLD (steatosis, inflammation and fibrosis) or the gut microbiome. As such, the gut microbiome presents an exciting new research field of which we have only scraped the surface. Unveiling how specific microbial subpopulations behave and change during NAFLD development might open the door to individualized treatment.

Several treatments have been tested in clinical trials, and whilst some promising results have been obtained, most have failed to deliver the desired outcome. The NAFLD treatment landscape is rapidly evolving as a consequence of our growing understanding of its underpinning mechanisms. Treatments aimed at ameliorating not one, but multiple, features of the condition hold great promise. Furthermore, our increasing appreciation the heterogeneity of the condition will enable us to develop more personalized therapies.

Whilst the holy grail has not yet been found; step by step, its quest is ongoing, and getting closer to the discovery of successful NAFLD treatments.

## Figures and Tables

**Figure 1 cancers-12-01714-f001:**
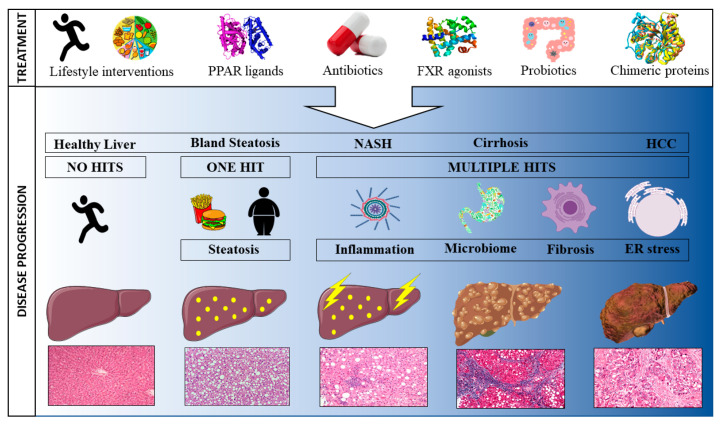
Schematic representation of the progression towards hepatocellular carcinoma and potential treatments to attenuate disease progression.

**Table 1 cancers-12-01714-t001:** Overview of current and pipeline treatments for NAFLD and NASH.

Target	Drug	Trial ID	BRCT	Cohort Medical Conditions	Ref.	Outcome
PPARα	Gemfibrozil	(Turkey)	Y	NASH	[34]	Decreased serum liver enzymes ^1^ and triglyceride
	Fenofibrate	(Spain)	N	NAFLD	[35]	Improved metabolic syndrome, decreased serum liver enzymes and triglycerides
	Clofibrate	(US – Pre-1997)	N	NASH	[36]	No improvement
	Omega-3 PUFA (Omacor)	Welcome – Phase IV, NCT00760513	Y	NAFLD	[38]	Decreased liver fat percentage
	Omega-3 PUFA (Omacor)	Phase III, NCT01277237	Y	NAFLD	-	TBD
PPARδ	Seladelpar (MBX-8025)± Atorvastatin	Phase II, NCT00701883	Y	Hyperlipidaemia	[41]	Decreased liver enzymes and improved serum lipid profile
	Seladelpar (MBX-8025)	Phase II, NCT03551522	Y	NASH	-	Trial Suspended – unexplained histological findings
PPARγ	Pioglitazone± vitamin E	PIVENS-Phase III, NCT00063622	Y	NAFLD, NASH	[46]	Reduced liver steatosis, lobular inflammation and serum ALT/AST
	Pioglitazone	UTHSCSA NASH-Phase IV, NCT00994682	Y	Type 2 Diabetes, NAFLD, NASH	[47]	Significant decrease in NAS score by ≥ 2 points in 58% participants; resolution of NASH in 51%.
	Lobeglitazone	ELLEGANCE - Phase IV, NCT02285205	N	Type 2 Diabetes, NAFLD	[48]	Improved liver and serum lipid profiles
PPARα/δ	Elafibranor (GFT505)	Phase IIa, NCT01271777	Y	Insulin resistance + abdominal obesity	[50]	Improved plasma lipids and hepatic insulin resistance, reduced liver inflammation and ALT
	Elafibranor (GFT505)	Phase II, NCT01271751	Y	Athero-genic dyslipidaemia + abdominal obesity	[50]	Decreased serum lipids and liver GGT
	Elafibranor (GFT505)	Phase II, NCT01275469	Y	Impaired glucose tolerance + abdominal obesity	[51]	Improved insulin sensitivity (HOMA-IR), fasting blood glucose and decrease in liver GGT
	Elafibranor (GFT505)	Phase IIb, NCT01694849	Y	NASH	[52]	No significant difference between placebo and elafibranor groups for primary outcome (resolution of NASH) ^2^
	Elafibranor (GFT505)	RESOLVE-IT – Phase III, NCT02704403	Y	NASH	-	Ongoing (recruiting)
PPARα/γ	Saroglitazar	EVIDENCES VI – Phase II, NCT03863574	Y	NASH	-	Ongoing (recruiting)
PPAR-pan	lanifibranor (IVA337)	NATIVE – Phase IIb, NCT03459079	Y	NAFLD, Type 2 Diabetes	-	Ongoing (recruiting)
Non-PPAR	Oral insulin (ORMD-0801)	Phase II, NCT02653300	N	NASH, Type 2 diabetes	-	Ongoing (recruitment)
	Liraglutide (GLP1 agonist)	LEAN-J	N	NASH	[61]	Decreased liver and visceral fat, liver enzymes and FPG
	Semaglutide (GLP1 agonist)	Phase II, NCT02453711	Y	Obesity, metabolic disorder	[62]	Decreased ALT and hsCRP, significant weight loss at all doses
	Semaglutide (GLP1 agonist)	SUSTAIN 6 – Phase III, NCT01720446	Y	Diabetes, Type 2 diabetes	[62]	Decreased ALT and hsCRP, decreased cardiovascular events (death, infarction or stroke)
	Armachol (SCD1 inhibition)	Aramchol003 - Phase II, NCT01094158	Y	NAFLD, NASH, Metabolic syndrome	[66]	Decrease in liver fat percentage at mid-dose
	Armachol (SCD1 inhibition)	ARMOR - Phase III/IV, NCT04104321	Y	NASH	-	Ongoing (recruiting)
	Dapagliflozin (SGLT2 inhibitor)	Dokkyo Medical University (Japan) - UMIN000022155	Y	NAFLD	[72]	Decreased liver fibrosis, visceral fat mass and liver enzymes
	Dapagliflozin (SGLT2 inhibitor) + omega-3 carboxylic acid	EFFECTII – Phase II, NCT02279407	Y	NAFLD, Type 2 diabetes	[73]	Significant reduction in liver fat and liver enzymes
	Dapagliflozin (SGLT2 inhibitor)	DEAN – Phase III, NCT03723252	Y	NASH	-	Ongoing (recruitment)
	Pentoxifylline (TNFα inhibitor)	Phase II, NCT00590161	Y	NASH	[78]	Improved liver steatosis, fibrosis and lobular inflammation
	Pentoxifylline (TNFα inhibitor)	(Sri Lanka) -SLCTR/2014/016	N	NASH	[79]	Lifestyle intervention+pentoxifyllin improved NAS
	Pentoxifylline (TNFα inhibitor)	Phase II/III, NCT00267670	Y	NASH	-	No difference between placebo and pentoxifylline groups
	Vitamin D3	Phase II, NCT01571063	Y	NASH	[87]	Decreased serum ALT
	Obeticholic acid (FXR1 ligand)	FLINT – Phase IIb, NCT01265498	Y	NASH, NAFLD	[98]	Improved liver NAS in 45% of patients, elevated pruritis
	Obeticholic acid (FXR1 ligand)	REGENERATE – Phase III, NCT02548351	Y	NASH	[100]	Improved fibrosis in 23% of patients, elevated pruritis
	NGM282 (FGF19 signalling)	Phase II, NCT02443116	Y	NASH	[104]	Reduction in liver fat content
	ND-L02-s0201 (Vitamin A-coupled siRNA to HSP47 – hepatic stellate cell fibrosis target)	METAVIR F3-4 - Phase Ib/II, NCT02227459	Y	Hepatic fibrosis	-	TBD
	Selonsertib (inhibitor of ASK1)	Multi-center Phase 2	N	NASH	[111]	Improved liver fibrosis
	Selonsertib (inhibitor of ASK1)	STELLAR 3 and 4 - Phase 3, NCT03053050 and NCT03053063	Y	NASH	[112]	No improvement in fibrosis, trial terminated
	Rifaximin (antibiotic)	Phase I, NCT02884037	Y	NAFLD, NASH	[123]	Reduction in proinflammatory cytokines, liver enzymes and NAFLD-liver fat score; improved insulin sensitivity (HOMA-IR)
	Rifaximin (antibiotic)	Phase II, EudraCT 2010–021515-17	N	NASH	[124]	Trial prematurely ended - no improvement
	VSL#3 (Probiotic)	VAIIO – Phase II, NCT01650025	Y	Obesity	[133]	Significant improvement in NAFLD

Trial ID, patient group and main outcomes are depicted. ALT: alanine aminotransferase; ASK1: apoptosis signal-regulating kinase 1; AST: alanine transaminase; BRCT: blinded, randomized, placebo-controlled trial; GGT: γ glutamyl transferase; hsCRP: high-sensitivity C-reactive protein; FXR: Farnesoid X receptor; HOMA-IR: homeostasis model assessment of insulin resistance; NAS: NASH activity score; stearoyl co-A desaturase: SCD1; TBD; results yet to be disclosed. ^1^ liver enzymes refer to hepatocyte injury biomarkers detected in the serum (for, e.g., ALT, AST, GGT, fibroblast growth factor 21, cytokeratin) ^2^ elafibranor resolved NASH in a greater proportion of patients with NAS score ≥ 4 than the placebo group, when a post-hoc analysis of the study was performed using a modified definition of NASH.

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
