# Peer review of "Current and Future Treatments in the Fight against Non-Alcoholic Fatty Liver Disease"

_cancers, 2020, doi:10.3390/cancers12071714_

Round 1
Reviewer 1 Report
The review is well written but could be more extensive in its consideration of the field; by example it has not addressed the other bile acid receptor TGR5; or compounds that target both TGR5 and FXR; and the family of G protein receptors, by example GRP84 that bind MCFAs and the compound PBI-4547 and others. This must be included.
Minor comments:
- The green-red background of figure 1 is not suitable for publication.
- Lines 231: correct to currently being examined.
- Line 260-262: is out of context with the above paragraph.
- Line: 309-311: is similarly out of context; make a related summary sentence
- Line 359-360: ditto as for points 3 and 4.
- At this stage Ref 111 is not on line and should be removed
Author Response
The review is well written but could be more extensive in its consideration of the field; by example it has not addressed the other bile acid receptor TGR5; or compounds that target both TGR5 and FXR; and the family of G protein receptors, by example GRP84 that bind MCFAs and the compound PBI-4547 and others. This must be included.
We would like to thank the reviewer for their time and comments provided. We have adjusted the manuscript according to the reviewer’s requirements and feel this has improved the current revised manuscript.
We agree with the reviewer that not every treatment is included in the current review. We have mainly focussed on treatments that are currently under investigation in phase 2 and 3 clinical trials, whilst still including some other treatments. We have included a section on the TGR5 bile acid receptor (line 346-356), GRP84 (line 296-301) and GPR13 (lines 449-452) as recommended by the reviewer. We hope this is to their satisfaction.
Minor comments:
- The green-red background of figure 1 is not suitable for publication.
The figure has been altered to a single coloured background.
- Lines 231: correct to currently being examined.
This has been corrected.
- Line 260-262: is out of context with the above paragraph.
We thank the reviewer for pointing this out. We have moved the sentence to the start of the paragraph (line 224-227).
- Line: 309-311: is similarly out of context; make a related summary sentence
We have altered the summary statement to better reflect the section (line 306-309)
- Line 359-360: ditto as for points 3 and 4
The summary statement has been amended to better reflect the above section (line 368-370).
- At this stage Ref 111 is not on line and should be removed
The study has since been accepted for publication. The reference has been changed to reflect this and is now Ref 115.
Reviewer 2 Report
In the present manuscript, Smeuninx et al. evaluated the proposed aetiology of NAFLD-related HCC, and summarised the potential therapeutic options for NASH-based HCC. While not entirely novel, the manuscript represents a massive effort of the authors and is thorough, with treatment options ranging from metabolic alterations to anti-inflammatory agents to probiotics. I myself very much enjoyed reading the paper.
Author Response
We thank the reviewer for their time and their kind comments. We are pleased to know the reviewer enjoyed reading the manuscript and agree the manuscript is not entirely novel. Whilst the NAFLD landscape is continuously evolving, we merely hope to have provided an updated summary on past, currently available and new treatments.
Reviewer 3 Report
To the authors of the manuscript,
It seems to be a misunderstanding about the terms of different pathologies and, as a consequence, the reading of the paper results quite difficult. The structure of the manuscript might also be improved.
My main concern is that you consider NAFLD as steatosis, whereas the term NAFLD is used to include the group of pathologies ranging steatosis, NASH and cirrhosis. Moreover, you also indicate that HCC is the final stage of NAFLD, and this condition is not included in the term.
Related to the structure, you focus so much on PPAR agonists that you divide the different therapies on PPPAR/non-PPAR. The PPAR-related part should be shorter in order to focus a little bit more about those "non-PPAR" therapies.You include gut-liver axis modulation in the chapter "Inflammation" that belongs to "Intrahepatic", and you also include gut-liver axis modulation in the chapter "Extrahepatic".
The part about lean NASH could be also reduced.
I really appreciate the table where you have included all the therapies.
Author Response
We would like to thank the reviewer for their time and insightful comments regarding the manuscript. We have amended the manuscript to reflect the proposed changes and hope this is to the satisfaction of the reviewer.
It seems to be a misunderstanding about the terms of different pathologies and, as a consequence, the reading of the paper results quite difficult. The structure of the manuscript might also be improved.
My main concern is that you consider NAFLD as steatosis, whereas the term NAFLD is used to include the group of pathologies ranging steatosis, NASH and cirrhosis. Moreover, you also indicate that HCC is the final stage of NAFLD, and this condition is not included in the term.
We fully agree with the reviewer and have amended the terminology throughout the paper to avoid confusion. In the first paragraph, lines 25-27 have been amended to better reflect NAFLD as the over-arching term including steatosis, NASH and cirrhosis. We have also amended NAFLD to bland steatosis in Figure 1.
Related to the structure, you focus so much on PPAR agonists that you divide the different therapies on PPPAR/non-PPAR. The PPAR-related part should be shorter in order to focus a little bit more about those "non-PPAR" therapies. You include gut-liver axis modulation in the chapter "Inflammation" that belongs to "Intrahepatic", and you also include gut-liver axis modulation in the chapter "Extrahepatic".
We thank the reviewer for their comment and agree the structure could be improved. It is, indeed, difficult to strictly divide treatments into those targeting intra- and extrahepatic factors as treatments in both categories might influence each other. We have therefore renamed section 2.1 as “Hallmark characteristics driving NAFLD” and 2.2 as “Gut microbiome”.
Furthermore, sections that were under “Inflammation” have been moved to “Antibiotics”. We have included non-PPAR therapies such as GPR agonists, TGR5 (lines 296-301 and 347-357) and GPR13 (lines 449-452). Whilst the PPAR related section is extensive, we feel it covers important therapies and would, hence, like to include it in the manuscript.
The part about lean NASH could be also reduced.
We have reduced the section on lean NASH to make it more succinct. We hope this is to the reviewer’s satisfaction.
I really appreciate the table where you have included all the therapies.
Thank you very much for your kind comments. We have updated the table to reflect the changed reference numbers.
Round 2
Reviewer 1 Report
It is much easier to read; there are still some typos and english grammar issues and these need to be touched on in the final read.
Author Response
We would like to thank the reviewer for their continued insights into the manuscript. Thank you for the positive comments and are glad the manuscript has been improved.
We have thoroughly read the paper for typos and English grammar. We hope to have found most, if not all, spelling errors.
Reviewer 3 Report
To the authors of the manuscript.
I appreciate the modifications realized in this new version. I have found it much more easier to read and enjoyed it.
I have added the comments in the PDF to make them easier to follow.
Best regards.

Author Response
We would like to thank the reviewer for their continued help, time and insights into the current manuscript. We are happy to hear the reviewer enjoyed the improvements made to the paper.
We have incorporated the suggestions made by the reviewer (red text colour) and hope this is to their satisfactions. We would also like to thank the reviewer for adding the comments into the PDF file, this was very helpful to amend the manuscript.
- we have amended the suggestions in the abstract.
- We have added the fibrosis in the NAS scoring system.
- We have further changed the terminology of NAFLD throughout the mansucript.
- We have added the VLDL reference
- The INT-787 drug is being tested in preclinical studies. This has been added.
- Dual PPAR agonists have been merged into one paragraph.
- The gut microbiome section has been moved to the discussion.